# CEGH: A Hybrid Model Using CEEMD, Entropy, GRU, and History Attention for Intraday Stock Market Forecasting

**DOI:** 10.3390/e25010071

**Published:** 2022-12-30

**Authors:** Yijiao Liu, Xinghua Liu, Yuxin Zhang, Shuping Li

**Affiliations:** 1School of Management Science and Engineering, Shandong University of Finance and Economics, Jinan 250014, China; 2Business School, Shandong University of Political Science and Law, Jinan 250014, China

**Keywords:** stock prediction, complete ensemble empirical mode decomposition, gated recurrent unit, attention, approximate entropy, sample entropy

## Abstract

Intraday stock time series are noisier and more complex than other financial time series with longer time horizons, which makes it challenging to predict. We propose a hybrid CEGH model for intraday stock market forecasting. The CEGH model contains four stages. First, we use complete ensemble empirical mode decomposition (CEEMD) to decompose the original intraday stock market data into different intrinsic mode functions (IMFs). Then, we calculate the approximate entropy (ApEn) values and sample entropy (SampEn) values of each IMF to eliminate noise. After that, we group the retained IMFs into four groups and predict the comprehensive signals of those groups using a feedforward neural network (FNN) or gate recurrent unit with history attention (GRU-HA). Finally, we obtain the final prediction results by integrating the prediction results of each group. The experiments were conducted on the U.S. and China stock markets to evaluate the proposed model. The results demonstrate that the CEGH model improved forecasting performance considerably. The creation of a collaboration between CEEMD, entropy-based denoising, and GRU-HA is our major contribution. This hybrid model could improve the signal-to-noise ratio of stock data and extract global dependence more comprehensively in intraday stock market forecasting.

## 1. Introduction

It is generally accepted that stock markets are crucial for modern societies and economies. With the increasing availability of stock time series at an intraday frequency and the development of quantitative trading, intraday stock market forecasting has become a hot issue for economists, investors, and regulators. Intraday stocks are securities that trade on the markets during regular business hours. Typically, 5-min, 30-min, and 60-min charts are used to capture intraday stock price movements. Unfortunately, intraday stock time series are noisier and more complex than other financial time series with longer time horizons. To begin with, the intraday market is more vulnerable to policy uncertainty and investor sentiment. Furthermore, the intraday time series contains intraday seasonal cycles. In addition, black swans, such as the current COVID-19 pandemic, can have a serious impact on stock markets, which are particularly sensitive to changes [1]. Moreover, the forecasting models often pick up noise instead of signals since intraday stock time series have low signal-to-noise ratios. These factors make it challenging to predict the intraday stock market. Several related methods have been proposed to predict the stock market. Traditional statistical models have a long history in stock forecasting. These models assume that time series are generated from a linear and stationary process and try to model the underlying time series generation process. However, this is inconsistent with the real stock market. The availability and analyzability of financial high-frequency data are gradually enhanced by the advantages of big data and artificial intelligence. Stock market forecasting has become an area at the intersection of finance and computer science. Machine learning has also drawn the attention of many researchers. Some classic machine learning models, such as the neural network (NN), support vector regression, and decision tree, have been applied for predicting stock time series [2]. Gu, Kelly, and Xiu predicted the stock return with various models and found that machine learning models significantly improved accuracy when compared with traditional statistical models [3].

In recent years, deep learning has been wildly used for stock prediction [4,5,6]. It extracts high-level features from a large amount of data by hierarchical hidden layers. Specifically, the recurrent neural network (RNN), one of the most popular deep learning models, has an internal memory that enables it to process sequences. Compared with the classic feedforward neural network (FNN), an RNN with memory cells takes information from prior inputs to influence the current input and output. Long short-term memory (LSTM) and gated recurrent unit (GRU) networks are variants of the RNN, and both are widely used to predict financial time series. Fischer and Krauss deployed LSTM for predicting stocks and found that LSTM outperformed memory-free methods [7]. Zhao et al. demonstrated that their GRU-based model performed slightly better than their LSTM-based model for stock price trend prediction [8]. Sako et al. used RNNs to predict the closing price of the stock index and also found that the GRU gave the overall best results [9]. It can be concluded that variants of the RNN have been the most frequently adopted deep learning models in stock prediction. However, RNNs still have some limitations, such as a lack of explanation and inefficiencies in learning sequential dependence.

Many researchers use attention models to narrow such gaps. Recently, the attention model has become an important concept and has been wildly used in NLP [10] and Computer Vision [11]. There are three neural architectures used in conjunction with attention [12]: (1) the encoder–decoder framework [13,14]; (2) the transformer [15,16]; and (3) memory networks [17]. In the financial field, researchers have applied attention to process text, such as economic news, and learn sequential representation for time series, such as price. To identify long-term temporal dependencies, Zhang et al. proposed an attention-enhanced LSTM model. The experimental results for stock price demonstrated that this attention-based model improved forecasting performance considerably [18]. Teng et al. proposed a multi-scale local cue and hierarchical attention-based LSTM model to capture stock price trend patterns [19]. Xu et al. predicted stock movements from financial news and historical prices using a GRU based on reinforcement learning with incorporated attention mechanisms [20]. Wang et al. proposed the AlphaStock investment strategy, using LSTM with history attention (LSTM-HA) to extract asset representation from multiple time series [21]. 

Although deep learning methods and attention mechanisms have improved feature representation, the complexity of stock data often leads to the risk of overfitting. Frequency decomposition methods are useful for reducing the complexity of stock time series. Empirical mode decomposition (EMD) can decompose any complicated data set into a finite and usually small number of intrinsic mode functions (IMFs) [22]. Unlike traditional Fourier spectral analysis, EMD does not require stationarity and linearity, although mode mixing occurs frequently with EMD. To overcome this drawback, ensemble empirical mode decomposition (EEMD), consisting of sifting and an ensemble of added white noise, has been proposed [23]. Complete ensemble empirical mode decomposition with adaptive noise (CEEMD) [24] can effectively eliminate residue noise in IMFs, which is an improvement on EEMD. These frequency decomposition methods could play a useful role in financial time series analysis if coupled with machine learning models. Cao et al. found that, compared with LSTM, the proposed CEEMDAN-LSTM model performed better in financial forecasting [25]. Zhang et al. proposed a hybrid CEEMD-PCA-LSTM model for stock prediction which outperformed benchmark models in terms of predictive accuracy and profitability performance [26]. Rezaei et al. predicted stock prices with a CEEMD-CNN-LSTM model which uses CEEMD to decompose stock price to different frequency spectra and uses CNN-LSMT to extract patterns and data dynamics [27]. Lv et al. proposed a hybrid model based on CEEMDAN, ADF, ARMA, and LSTM to predict the stock index [28]. The results of these studies suggest that the hybrid models based on decomposition and RNNs perform better than the individual models in stock forecasting. Improving the performance of hybrid models using various methods is a rising trend in stock prediction.

Intraday stock time series are inherently noisy, and therefore need denoising before forecasting. Information entropy can measure the complexity of time series, which helps to filter noise. Delgado-Bonal and Marshak showed that approximate entropy (ApEn) and sample entropy (SampEn) are two algorithms for determining the regularity of series based on the existence of patterns [29]. Considering the temporal dimension of uncertainty, Vinte and Ausloos proposed a comprehensive cross-sectional volatility estimator for stock markets based on intrinsic entropy [30]. Olbrys and Majewska used SampEn to capture sequential regularity in stock market time series [31]. Raubitzek and Neubauer found a correlation between ApEn, SampEn, and predictability in stock market data [32]. Moreover, some studies have introduced entropy to the EMD models. Chou et al. extracted the entropy features (ApEn and SampEn) of EMD-derived signals to predict the fall risk of elderly people [33]. Shang et al. combined the advantages of CEEMD and ApEn to eliminate the influence of noise in partial discharge and demonstrated the hybrid model’s effectiveness and superiority [34]. 

Despite the extensive studies above, using hybrid models combining frequency decomposition methods and machine learning for intraday stock market forecasting still faces several limitations. First, there has been little discussion about the combination of entropy and EMD-based hybrid models in stock forecasting. Specifically, very little attention has been paid to filtering the high-frequency noise effectively after decomposition for stock forecasting. Second, most hybrid models use the same forecasting model for the IMFs indiscriminately, ignoring the characteristics of different signals. Third, most existing RNN models only use RNN layers to learn the sequential dependence in hidden states or directly stack the hidden states as inputs of the next layer. These models cannot effectively learn the global dependence of stock time series.

To overcome such limitations, we propose a CEGH (CEEMD-Entropy-GRU-HA) model for intraday stock market forecasting. This hybrid model is composed of frequency decomposition, entropy, GRU, and attention mechanisms. We first use CEEMD to decompose the original intraday stock market data into IMFs. Then, entropies, which can measure the complexities of signals, are used to denoise the IMFs. On this basis, the retained IMFs are divided into several groups and predicted using the FNN or the GRU with history attention (GRU-HA). Finally, we integrate the prediction results of the groups to obtain the ensemble results.

The main contributions are summarized as follows: (1) To the best of our knowledge, our work is among the first to attempt to create a collaboration between frequency decomposition, entropy, and attention. The proposed model was compared with the individual state-of-the-art models and other decomposition-based models. The experiments on the U.S. and China markets showed that this hybrid model improved accuracy in intraday stock forecasting; (2) we introduced an attention mechanism to IMF forecasting, which enhanced the ability to explore global dependence in stock time series; (3) we used entropies to remove the noise with high complexity, and this could improve the signal-to-noise ratio of the intraday stock data.

The rest of this paper is organized as follows. Section 2 describes the proposed CEGH model. Section 3 describes the experiments on the U.S. stock market and the China stock market. Section 4 evaluates the forecasting performance of the CEGH model and discusses the findings. Section 5 concludes this paper. 

## 2. The CEGH Model

In this section, we first introduce the framework of CEGH in general and then describe the four main components.

### 2.1. Framework

To improve the signal-to-noise ratio of intraday stock data and extract global dependence more comprehensively, we propose a hybrid CEGH model. This model combines frequency decomposition, entropies, recurrent neural networks, and attention mechanisms. As is shown in Figure 1, the model consists of four stages.

Stage 1: Decomposition. Intraday forecasting with raw stock data is challenging due to the complexity of financial time series. Using highly complex data may result in the poor predicting performance of an excessively complicated machine learning model. To better extract high-level features from the original intraday stock data, CEEMD is used to decompose the raw data into several components with a well-defined instantaneous frequency using IMFs.

Stage 2: Entropy-based denoising. Information entropy provides information regarding the complexity of the time series, which helps to filter the noise. In this stage, the approximate entropy and sample entropy of each IMF are calculated, and a certain noise threshold is set to filter the noise.

Stage 3: Grouping and Forecasting. Different IMFs have different complexities and time–frequency characteristics which need different prediction models. In this stage, the retained IMFs are divided into a high-frequency group, a medium-frequency group, a low-frequency group, and a trend group. The comprehensive signals of each group are then predicted using the FNN or the GRU-HA separately.

Stage 4: Ensemble. The final prediction results are obtained by integrating the prediction results for each group in Stage 3.

### 2.2. CEEMD

In Stage 1, CEEMD [24] is used to decompose the original intraday stock data into IMFs. CEEMD is a noise-assisted EMD technique, which can be described with the following steps.

Step 1: Add white noise to the original intraday stock data x as follows:(1)$$x^{i}=x+\varepsilon_{0}w^{i},$$
where $w^{i}(i=1,2,\ldots ,I)$ are different realizations of white noise and $\varepsilon_{0}$ is the parameter of white noise power.

Step 2: Decompose $x^{i}$ using EMD to obtain their first modes. By averaging $IMF_{1}^{i}$, the first component can be obtained as follows:(2)$$IMF_{1}=\frac{1}{I}\sum_{i=1}^{I}IMF_{1}^{i}.$$

Step 3: Calculate the first residue as follows: (3)$$r_{1}=x-IMF_{1}.$$

Step 4: Calculate the second IMF as follows: (4)$$IMF_{2}=\frac{1}{I}\sum_{i=1}^{I}E_{1}\left(r_{1}-\varepsilon_{1}E_{1}\left(w^{i}\right)\right),$$
where $E_{j}(\cdot )$ is the operator that produces the *j*-th mode obtained using EMD.

Step 5: For k=2,…,K, calculate the *k*-th residual as follows:(5)$$r_{k}=r_{k-1}-IMF_{k}.$$

Step 6: Calculate the (*k* + 1)-th IMF as follows:(6)$$IMF_{k+1}=\frac{1}{I}\sum_{i=1}^{I}E_{1}\left(r_{k}+\varepsilon_{k}E_{k}\left(w^{i}\right)\right).$$

Step 7: Repeat Step 5 and Step 6 until the obtained residual can no longer be decomposed feasibly. The final residual can be described as follows:(7)$$r=x-\sum_{k=1}^{K}IMF_{k}.$$

Therefore, the given intraday stock data can be expressed as follows: (8)$$x=\sum_{k=1}^{K}IMF_{k}+r.$$

### 2.3. Entropy-Based Denoising

In Stage 2, information entropy is used to remove those IMFs that represented noise components. Because of the low signal-to-noise ratios of intraday stock data, denoising is particularly important. Information entropy is utilized to quantify the complexity and regularity of time series. A larger value means that the sequence has a higher probability of generating a new pattern. The larger the information entropy value, the more complex and irregular the sequence is. Unlike previous studies that used only single entropy for denoising [34], we use approximate entropy (ApEn) [35] and sample entropy (SampEn) [36] to measure the complexities of intraday stock IMFs.

In our model, if the entropy value is above a certain threshold, the IMF is regarded as noise and is discarded. Otherwise, the IMF is assumed to contain useful intraday stock market information and is kept. Given a time series u(i),i=1,2,…,N, we define an embedding dimension m and a tolerance r. The calculation of ApEn and SampEn can be defined with the following steps.

Step 1: Extend u(i) to the $m^{th}$ vector $V_{m}(i)$ as follows:(9)$$V_{m}\left(i\right)=\left[u\left(i\right),u\left(i+1\right),\ldots ,u\left(i+m-1\right)\right],$$
where i=1,2,…,N−m+1.

Step 2: Calculate the distance between $V_{m}\left(i\right)$ and $V_{j}\left(j\right)$ as follows:(10)$$D\left[V_{m}\left(i\right),V_{m}\left(j\right)\right]=\underset{k=0,1,\ldots ,m-1}{\max}\left\{\left|u\left(i+k\right)-u\left(j+k\right)\right|\right\},$$
where j=1,2,…,N−m+1,j≠i.

Step 3: Calculate approximate entropy. First, measure the regularity and frequency of patterns within tolerance r as follows:(11)$$C_{i}^{m}(r)=\frac{\mathrm{number}\;\mathrm{of}\;j\;\mathrm{such}\;\mathrm{that}\;D\left[V_{m}\left(i\right),V_{m}\left(j\right)\right]\leq r}{N-m+1}.$$
Then, calculate the mean value of the logarithm of $C_{i}^{m}\left(r\right)$ as follows:(12)$$\varphi^{m}\left(r\right)=\frac{\sum_{i=1}^{N-m+1}\ln\left[C_{i}^{m}\left(r\right)\right]}{N-m+1}.$$
Finally, the ApEn can be defined as follows:(13)$$ApEn\left(m,r\right)=\varphi^{m}\left(r\right)-\varphi^{m+1}\left(r\right).$$

Step 4: Calculate sample entropy. First, calculate the two coefficients $A^{m}\left(r\right)$ and $B^{m}\left(r\right)$ as follows:(14)$$A_{i}^{m}\left(r\right)=\frac{\sum_{j=1,j\neq i}^{N-m}\mathrm{number}\;\mathrm{of}\;\mathrm{times}\;\mathrm{that}\;D\left[V_{m+1}(i),V_{m+1}\left(j\right)\right]<r}{N-m-1},$$
(15)$$B_{i}^{m}\left(r\right)=\frac{\sum_{j=1,j\neq i}^{N-m}\mathrm{number}\;\mathrm{of}\;\mathrm{times}\;\mathrm{that}\;D\left[V_{m}\left(i\right),V_{j}\left(j\right)\right]<r}{N-m-1}.$$
Add them as follows:(16)$$A^{m}\left(r\right)=\frac{\sum_{i=1}^{N-m}A_{i}^{m}\left(r\right)}{N-m},$$
(17)$$B_{N}^{m}\left(r\right)=\frac{\sum_{i=1}^{N-m}B_{i}^{m}\left(r\right)}{N-m}.$$
The SampEn can be defined as follows:(18)$$SampEn\left(m,r\right)=-\ln\left[\frac{A^{m}\left(r\right)}{B^{m}\left(r\right)}\right].$$

### 2.4. GRU-HA

After decomposing and grouping, we use different models to predict different group signals. The classic FNN is used for the relatively regular groups and the GRU-HA is used for the more complex groups. The GRU-HA combines the GRU and an attention mechanism as shown in Figure 2. First, the GRU layers learn the hidden states of the input stock data, and then the history attention layers exploit the global dependency in the hidden states. The history attention representation and the last hidden state are concatenated into an input vector of a fully connected layer. Finally, we obtain the output of the dense layer as the predicted value. The main components of the GRU-HA are described in detail as follows.

#### 2.4.1. GRU

With the GRU-HA, we first use the GRU to learn the history state of the input data as follows:(19)$$h_{t}=GRU\left(h_{t-1},x_{t}\right),t\in \left[1,T\right],$$
where $x_{t}$ denotes the total signal of each group at time step t and $h_{t}$ denotes the hidden state, which contains the sequential dependence, leaned by the GRU. The structure of the GRU is shown in Figure 3. 

At the time step t, the activation $h_{t}^{j}$ of the j-th GRU unit is a linear interpolation between the previous activation $h_{t-1}^{j}$ and the candidate activation ${\tilde{h}}_{t}^{j}$:(20)$$h_{t}^{j}=\left(1-z_{t}^{j}\right)h_{t-1}^{j}+z_{t}^{j}{\tilde{h}}_{t}^{j},$$
where the update gate $z_{t}^{j}$ controls how much the unit updates its information. The update gate is defined as follows:(21)$$z_{t}^{j}=\sigma {\left(W_{z}x_{t}+U_{z}h_{t-1}\right)}^{j}.$$
The candidate activation ${\tilde{h}}_{t}^{j}$ is calculated as follows:(22)$${\tilde{h}}_{t}^{j}=\tanh{\left[Wx_{t}+U\left(r_{t}⊙h_{t-1}\right)\right]}^{j},$$
where $r_{t}$ is the reset gate which is defined as follows:(23)$$r_{t}^{j}=\sigma {\left(W_{r}x_{t}+U_{r}h_{t-1}\right)}^{j}.$$

#### 2.4.2. History Attention

The hidden state $h_{t}$ learned by the GRU can represent the sequential dependencies of $x_{t}$, but cannot effectively learn the global dependencies. To enhance the efficiency of dependency learning, we combine the GRU with history attention.

We define a query vector $q_{T}$ for the last hidden state $h_{T}$:(24)$$q_{T}=W_{q}h_{T}.$$
We then define a key vector $k_{t}$ and a value vector $v_{t}$ for all hidden states $h_{t}$:(25)$$k_{t}=W_{k}h_{k},$$
(26)$$v_{t}=W_{v}h_{t},$$
where $W_{q}$, $W_{k}$, $W_{v}$ are the parameters to learn. Then, following Luong’s multiplicative style, the attention score is computed as follows: (27)$$\alpha_{t}=q_{t}k_{t}^{T}.$$
Next, the attention scores are used to calculate the attention weight: (28)$$ATT\left(q_{T},k_{t}\right)=\frac{\exp\left(\alpha_{t}\right)}{\sum_{i=1}^{T}\exp\left(\alpha_{i}\right)}.$$
Finally, the history states attention representation can be calculated as follows:(29)$$r=\sum_{t=1}^{T}ATT\left(q_{T},k_{t}\right)v_{t}.$$

## 3. Experiments

In this section, we empirically evaluate the proposed model using the U.S. stock market and China stock market. All the experiments are performed in Python 3.6.12. Specifically, the experiments related to neural network models and attention models are implemented using the deep learning end-to-end platform Tensorflow. The decomposition models are implemented using PyEMD and pyhht.

### 3.1. Data

The U.S. stock market data were collected from the Trade and Quote database (TAQ) of Wharton Research Data Services (WRDS). The Standard and Poor’s 500 (SP500) was chosen to represent the U.S. stock market. We used the price of SPY, the actively traded SP500 ETF, to represent the SP500. The U.S. stock market opens at 9:30 and closes at 16:00 Eastern Time. Every trading day has 13 half-hour intervals. The China stock market data were obtained from the Wind database. The China Securities Index 300 (CSI300) typically represents the overall China stock market. Chinese stock exchanges operate from 9:30 to 11:30 and from 13:00 to 15:00. Every trading day has eight half-hour intervals. The sample period spans from January 2015 through December 2020, a period which covers the well-known market event, the COVID-19 pandemic. The data are collected every half-hour during stock market trading hours.

We selected the last 20% of the sample for testing (out-of-sample data) and split the remaining data (in-sample data) into a training set and a validation set for 9:1.

### 3.2. Decomposing and Denoising

As mentioned before, we used CEEMD to decompose the raw intraday stock market signals into IMFs. The number of trials was set to 200, and the white noise standard deviation was set to 0.2. The decomposition results are shown in Figure 4. The arrangement of IMFs is from high to low frequency, and the residual occurs at the end.

To improve the signal-to-noise ratio of the intraday stock data, we use entropy-based denoising to remove the irregular noise. According to previous research [34], the parameters of the entropy algorithms are defined as: m=2 and $r=0.2E_{SD}$, where $E_{SD}$ is the standard deviation of the IMFs. The ApEn values and SampEn values of the IMFs are shown in Table 1. Different IMFs process different information entropy values, which means that different degrees of complexity exist in the diverse decomposition levels. The ApEn and SampEn values are gradually reduced for both the SP500 or CSI300, so the irregularities of the IMFs are gradually decreased. We set the noise thresholds as $\lambda_{ApEn}=1$ and $\lambda_{SampEn}=0.6$ for the two entropies. The decomposed sub-signals can be retained only if the ApEn value is less than 1 and the SampEn value is less than 0.6. Then, IMF1 and IMF2 are abandoned as noise. IMF3–IMF11 are kept as useful information for training.

### 3.3. Grouping and Forecasting

After denoising, we divide the retained IMFs into several groups and use different models to predict them.

#### 3.3.1. Grouping

The IMFs are grouped based on their characteristics. The standard deviation values are shown in Table 1. For the U.S. stock market, the standard deviations of IMF3 and IMF4 are relatively small, both less than 0.5. This reflects the fluctuations caused by short-term news. Therefore, IMF3 and IMF4 are used as the high-frequency group. The amplitudes of IMF5–IMF8 increase gradually with the standard deviation concentrated in the 0.7–2.5 range, and the period of IMF5–IMF8 becomes longer, reflecting the fluctuations caused by medium-term market factors. Therefore, IMF5–IMF8 are regarded as the medium-frequency group. The standard deviations of IMF9 and IMF10 are close to each other, around 5.7, and the periods become longer. This represents fluctuations caused by special events, such as the COVID-19 pandemic. Therefore, IMF9 and IMF10 are regarded as the low-frequency group. IMF11, the residual term after decomposition, represents the long-term economic situation of the U.S. stock market and is regarded as the trend group. As with the U.S. stock market, the retained IMFs of the China stock market are divided into four groups. IMF3, IMF4–IMF7, IMF8–IMF10, and IMF11 are regarded as the high-frequency group, the medium-frequency group, the low-frequency group, and the trend group, respectively.

The different groups reflect the volatility of the stock market in different financial cycles. For the U.S. stock market, the fourth part of Figure 5a shows that there is an overall growth trend during 2015–2019, but that the growth rate slows down significantly in 2018 and 2019. As can be seen in the third part, there is a relatively stable trend from 2015 to 2017. The market fluctuates greatly in 2018 and 2019. In 2020, the market resembles a rollercoaster, influenced by the COVID-19 pandemic. The first and second parts of Figure 5a show that there were extreme short-term fluctuations in 2020.

For the China stock market, as is shown in the fourth part of Figure 5b, the stock market has been on an upward trend from 2015 to 2019. Compared with the U.S. stock market, the China stock market displays a steeper curve, reflecting the rapid growth of the Chinese capital market. The fluctuations of the low-frequency group reflect the influences of long-term market factors. As illustrated in the third part of Figure 5b, the stock market in 2015–2016 experiences a rapid growth followed by a huge recession, which corresponds to the stock market crash in 2015. Moreover, the first and second parts of Figure 5b show that this recession is accompanied by intense fluctuations, and the rise in fluctuations was due to the government’s bailout policy, which did not reverse the bear market trend. This decline was accompanied by the participation of a large amount of leveraged funds, which also led to a greater fluctuation and a stronger downtrend.

#### 3.3.2. Forecasting

Before training, the input data had to be rescaled within the range of 0 and 1. We use the min–max scaler to normalize the grouped IMFs as follows:(30)$$X_{in}=\frac{X-\min\left(X\right)}{\max\left(X\right)-\min\left(X\right)}.$$
After predicting, the output values were converted back to their original scale by reversing Equation (30).

Table 2 shows the hyperparameters of the forecasting models. As mentioned before, we select different models to predict groups by their characteristics. Apparently, predicting a complex group is much more difficult than predicting a relatively regular one. As is shown in Figure 1, the high-frequency group and the medium-frequency group are predicted using the GRU-HA, and the low-frequency group and the trend group are predicted using the FNN. The FNN is composed of dense layers, i.e., regular densely connected neural network layers. The GRU-HA consists of GRU layers, attention layers, and dense layers. The grid search method is employed to determine the appropriate network structure, including the number of hidden layers and the number of units. For the hidden layers, we apply the rectified linear unit (ReLU) activation function to obtain the maximum of 0 and the input tensor as follows:(31)ReLU(x)=max(0,x).

For training, we use Adam [37], the stochastic gradient descent method based on adaptive moment estimation, for optimization. Its default learning rate is set to 0.001. In terms of loss function, we use the mean squared error (MSE) to compute the mean of squares of errors between labels and predicted values as follows:(32)$$MSE=\frac{1}{m}\sum_{i=1}^{m}{\left({\hat{y}}_{i}-y_{i}\right)}^{2}.$$

The number of epochs is the number of complete passes through the entire training set. Since too many epochs could lead to overfitting, we use early stopping as regularization. A training loop will check at the end of every epoch whether the loss is no longer decreasing. The maximum number of epochs is set to 500. 

## 4. Results and Discussion

To test the forecasting performance of our model on the intraday stock market, we conducted experiments using the proposed CEGH model and the baseline methods. In this section, we first define the evaluation metrics for the forecasting models. The baseline methods are then described. Last, we report the results and discuss the findings in two phases: group results and ensemble results. 

### 4.1. Evaluation Metric

The evaluation metric indicates the difference between the predicted value and the actual value. In our study, mean absolute error (MAE) and root mean squared error (RMSE) are used as metrics to evaluate the forecasting models, and they can be defined as follows:(33)$$MAE=\frac{1}{n}\sum_{i=1}^{n}\left|{\overset{⌢}{y}}_{i}-y_{i}\right|,$$
(34)$$RMSE=\sqrt{\frac{1}{n}\sum_{i=1}^{n}{\left({\hat{y}}_{i}-y_{i}\right)}^{2}},$$
where ${\hat{y}}_{i}$ is the predicted value and $y_{i}$ is the actual value. The lower the above two error values, the better the forecasting performance.

### 4.2. Baseline Methods

The proposed CEGH model is compared with some baseline models, including:

Feedforward neural network (FNN): the classical artificial neural network in which the connections of nodes do not form a loop.

Long short-term memory (LSTM) [38]: a wildly used extension of the recurrent neural networks (RNN) with three logic gates.

Gated recurrent unit (GRU) [39]: a variant of LSTM with a simpler unit structure.

GRU-HA: a hybrid model that combines a GRU and history state attention.

CEEMD-GRU-HA: a hybrid model using CEEMD, a GRU, and HA. Compared with CEGH, CEEMD-GRU-HA does not have an entropy-based denoising sector.

The FNN, LSTM, GRU, and GRU-HA models are undecomposed models. The CEEMD-GRU-HA model is a decomposition-base model. Specifically, the GRU-HA and CEEMD-GRU-HA models are proposed in this paper.

### 4.3. Group Results

As mentioned in Section 3.3 we divided the IMFs into a high-frequency group, a medium-frequency group, a low-frequency group, and a trend group. After that, the total signal of each group was predicted. Using the method described above, we obtained the group results. 

Table 3 shows the performance comparison for the groups in stage 3. The control group was composed of IMF1, IMF2, and IMF3. The different total signals of the groups have different characteristics and degrees of complexity. Training the models on the groups with low complexity is easier than training them using the IMFs with high complexity. For the control group, the high-frequency group, and the medium-frequency group, we used the FNN, LSTM, and GRU models as baseline methods. The trend group and low-frequency group are predicted directly using the FNN due to the relatively low complexity of their patterns. Some interesting aspects of this table are highlighted below.

(1) The recurrent neural network models performed better than the FNN, a finding which is consistent with results obtained in previous studies [8]. For the high-frequency U.S. stock market group, the MAE values of the FNN, LSTM, and GRU models were 0.0774, 0.0312, and 0.0308, respectively. For the high-frequency China stock market group, the MAE values of the FNN, LSTM, and GRU models were 1.1830, 0.6829, and 0.4744, respectively. These results indicate that for the two RNN variants, the GRU model performed better than LSTM model, a finding which is consistent with a previous study [9]. It may be reasonable to suppose that GRU is a better choice when the size of training sample is limited. 

(2) The GRU-HA performed better than the GRU. For the high-frequency U.S. stock market group, the MAE value of the GRU-HA was 18% lower than that of the GRU. The MAE value of the GRU-HA was as much as 80% lower than that of the GRU for the medium-frequency group. For high-frequency China stock market group, the MAE value of the GRU-HA was 7% lower than that of the GRU. The MAE value of the GRU-HA was 35% lower than that of the GRU for the medium-frequency group. A similar descending trend was also true for the RMSE value. These results imply that adding attention to the GRU enhances the prediction performance. Moreover, the performance improvement of the medium-frequency group was much larger than that of the high-frequency group. The high-frequency group retains only IMF3 after denoising. The GRU is sufficient for identifying this sequential pattern of relatively low complexity. The medium-frequency group composed of multiple IMFs is more complex. It may be inferred that the GRU-HA has more significant advantages for complex time series prediction.

### 4.4. Ensemble Results

In our study, the ensemble results were obtained by integrating the prediction results of the high-frequency group, the medium-frequency group, the low-frequency group, and the trend group, as explained in the previous subsection. The baseline method, CEEMD-GRU-HA, integrated the results of the above groups. The FNN, LSTM, GRU, and GRU-HA models were undecomposed models, as the original intraday stock data were predicted by these models directly.

Table 4 shows the performance comparison of the CEGH and baseline methods for the intraday stock market. In general, the proposed model had the lowest forecasting error, which verified its effectiveness. Some interesting observations are highlighted below.

(1) Specifically, the decomposition-based models outperformed the undecomposed models, which is consistent with results obtained in previous studies [27,40,41]. For U.S. stock market, the MAE values of the decomposition-based models were about 20% lower than that of the best-performing undecomposed model, the GRU-HA. For China stock market, the MAE values of the decomposition-based models were about 40% lower than that of the best-performing undecomposed model, the GRU-HA. The RMSE values exhibited a similar descending trend. These results indicate that such hybrid methods could extract the sequential representation more efficiently. A possible explanation for this is that the decomposition method could transform the original intraday stock time series into several relatively regular components. Apparently, time series with low complexity are much easier to predict.

(2) According to the results for the undecomposed models in Table 4, the GRU-HA had the lowest error values, which is consistent with the group results in Section 4.3. The results indicate that the RNN models performed better than the FNN. They also imply that the idea of introducing history attention to the GRU for intraday stock market forecasting is effective. Compared with the existing models that only use RNNs to learn the sequential dependence in stock time series, the GRU-HA extracted global dependence more comprehensively.

(3) According to the results for the decomposition-based models in Table 4, the CEGH outperformed the CEEMD-GRU-HA. As can be seen, the MAE errors of the CEGH were approximately 7% lower than those of the CEEMD-GRU-HA for both the U.S. and China stock markets. Specifically, from the comparison of the control group and the high-frequency group in Table 3, it can be seen that the errors of the high-frequency group were much lower than that of the Control group. This could be attributed to the entropy-based denoising section of the CEGH. ApEn and SampEn can measure the complexity of stock time series to recognize the noisy components in IMFs. It is an effective way to improve forecasting performance because it allows us to increase the signal-to-noise ratio before training the model. 

Furthermore, Figure 6 shows brief graphs of the ensemble results for a better comparison. In these figures, the blue lines represent the actual values, the red lines indicate the predicted values, and the green lines express the errors between the actual values and the predicted values. As we can see, the proposed CEGH model had a poor performance during the COVID-19 outbreak. Figure 6a indicates that the absolute values of the errors became significantly larger from February to March 2020. During this time, the U.S. stock market fell sharply due to the impact of the COVID-19 pandemic. As we can see in Figure 6b, the absolute values of the errors were extremely large in early February. For the China stock market, the errors were highly volatile when the market fell again around March 2020. It is noteworthy that the stock market was closed due to the Chinese New Year holiday at the end of January. This may imply that the proposed model is not good at learning from the limited historical data of intraday stock markets under the influence of black swan events. 

## 5. Conclusions 

We proposed a hybrid CEGH model to better analyze and predict intraday stock markets. This model combined frequency decomposition, entropy, GRU, and history attention. Based on the results and discussion of experiments on the U.S. and China stock markets, the following conclusions can be drawn: (1) The decomposition-based models outperformed the undecomposed models, demonstrating the advantages of CEEMD. This could be due to the fact that CEEMD reduced the complexity of the intraday stock time series. (2) The GRU-HA had a better performance than other individual models. This implies that introducing history attention to the GRU for intraday stock market forecasting is effective, as the history attention could allow it to extract global dependence more comprehensively. (3) The prediction error of the CEGH is smaller than that of the CEEMD-GRU-HA. This indicates that the entropy-based denoising section of the CEGH improved the forecasting performance. An explanation could be that the ApEn and SampEn measured the complexity of the stock time series to recognize the noisy components in IMFs, which allowed us to increase the signal-to-noise ratio before training the model. (4) The CEGH performed poorly at the beginning of the COVID-19 pandemic since it is difficult to learn patterns from limited historical training data during black swan events. This tells us that machine learning forecasting models should not be blindly relied on when extreme events occur in the stock market.

Overall, our results provide compelling evidence for the effectiveness of the CEGH in intraday stock market forecasting. However, some limitations are worth noting: (1) the stock forecasting model was only evaluated using error values; (2) the model did not take into consideration the fact that the stock market changed its behavior due to the economic situation; (3) the experiments were only conducted on the U.S. and China stock markets. Future works could be investigate the following: (1) incorporating the trading strategy and the profit evaluation; (2) combining adaptive learning methods and trading rules based on financial concepts; (3) comparing European or other conditions.

## Figures and Tables

**Figure 1 entropy-25-00071-f001:**
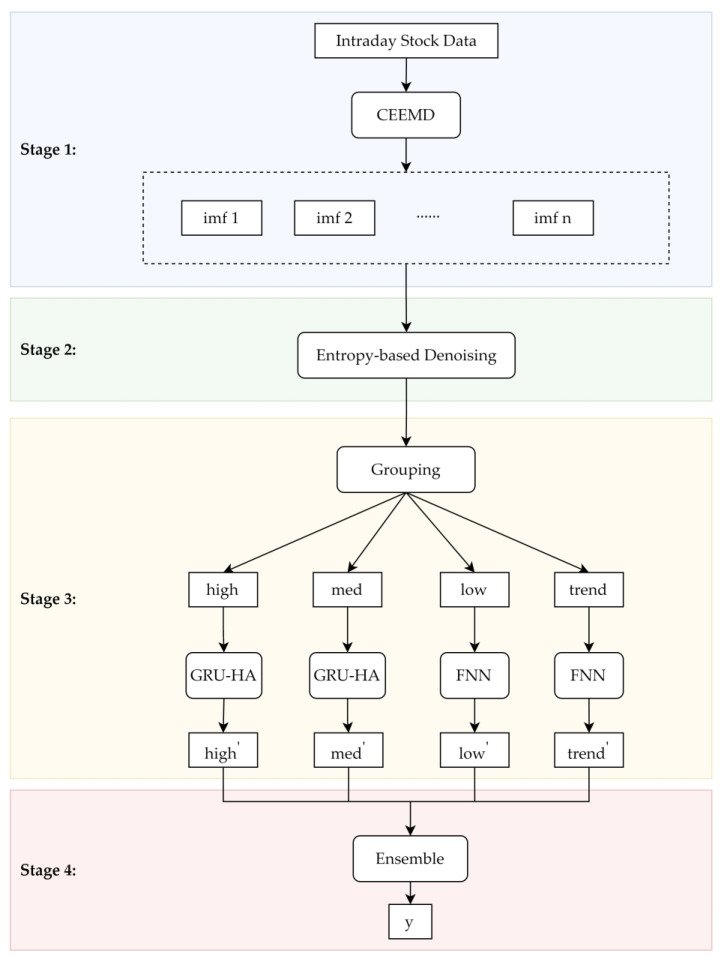
The framework of CEGH.

**Figure 2 entropy-25-00071-f002:**
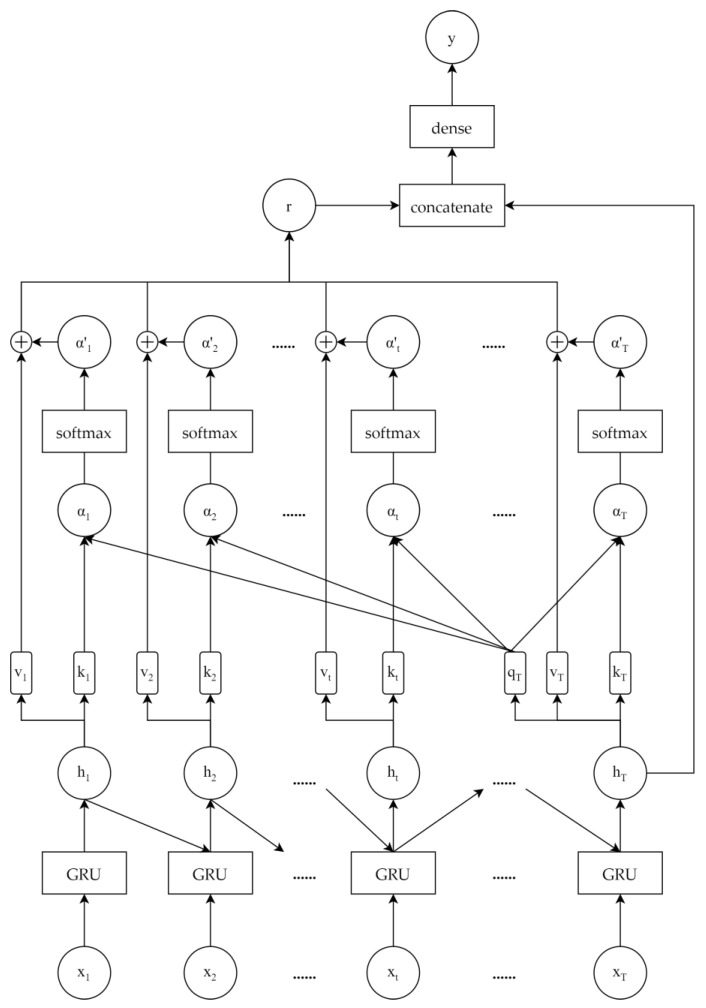
The structure of the gate recurrent unit with history attention (GRU-HA).

**Figure 3 entropy-25-00071-f003:**
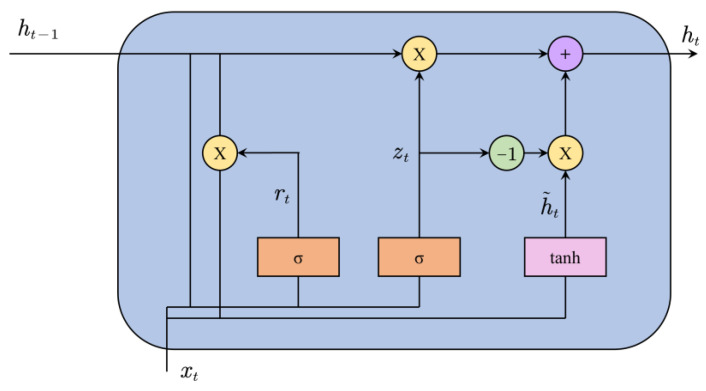
The structure of the GRU.

**Figure 4 entropy-25-00071-f004:**
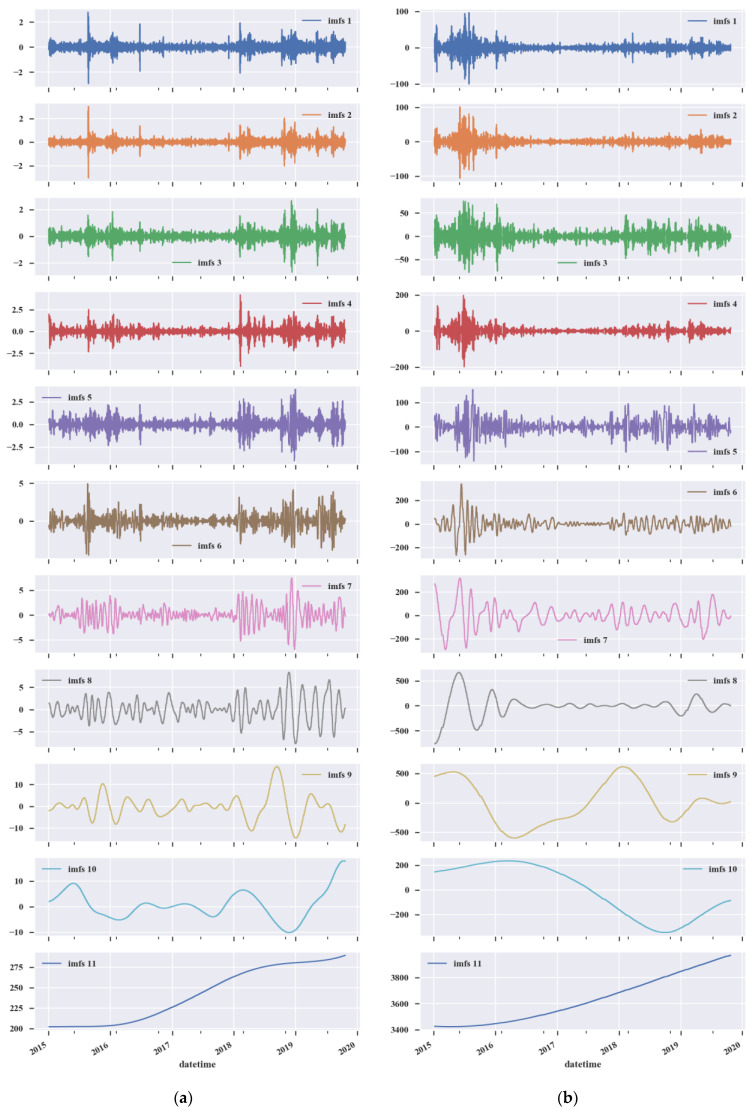
The CEEMD decomposition results. (**a**) The IMFs of the U.S. stock market; (**b**) the IMFs of the China stock market.

**Figure 5 entropy-25-00071-f005:**
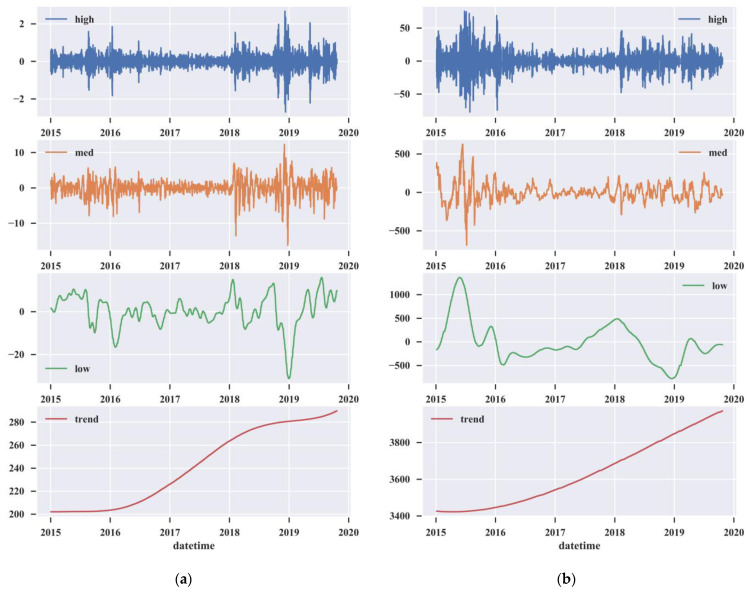
The total signals of the groups. (**a**) U.S. stock market; (**b**) China stock market.

**Figure 6 entropy-25-00071-f006:**
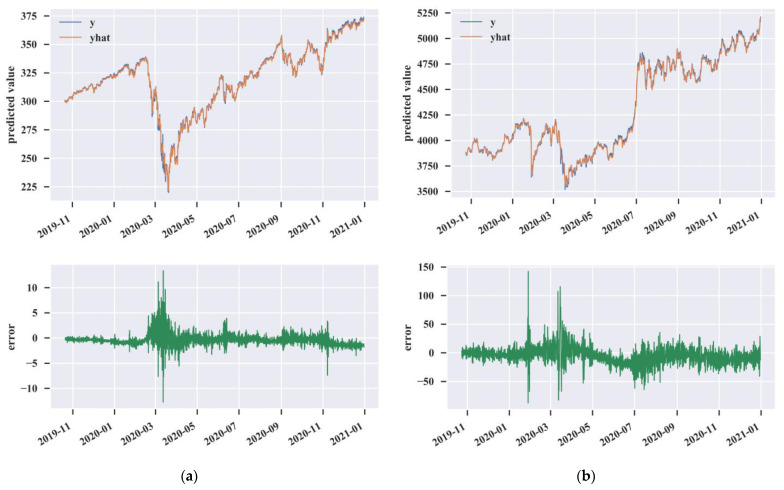
The ensemble results and prediction errors. (**a**) U.S. stock market. (**b**) China stock market.

**Table 1 entropy-25-00071-t001:** ApEn, SampEn, and standard deviations of the IMFs.

IMF	SP500			CSI300		
	ApEn	SampEn	std	ApEn	SampEn	std
IMF1	1.5097	1.0883	0.2128	1.3405	1.0185	9.5485
IMF2	1.2393	0.9526	0.2224	0.9256	0.6293	9.9564
IMF3	0.6819	0.5577	0.3087	0.6645	0.5572	13.3986
IMF4	0.5909	0.4734	0.4841	0.5105	0.3777	25.4280
IMF5	0.4468	0.3271	0.7496	0.3611	0.2668	32.2427
IMF6	0.2071	0.1546	1.0234	0.1132	0.0863	56.3356
IMF7	0.0758	0.0646	1.6871	0.0419	0.0370	91.7791
IMF8	0.0316	0.0293	2.5416	0.0120	0.0069	206.8343
IMF9	0.0089	0.0076	5.7662	0.0036	0.0032	358.5965
IMF10	0.0029	0.0028	5.6095	0.0010	0.0009	207.4260
IMF11	0.0001	0.0002	32.6616	0.0001	0.0003	175.7148

**Table 2 entropy-25-00071-t002:** Hyperparameter settings of the forecasting models.

Parameters	Values
Types of layers	Dense, GRU, Attention
Time steps	40
Number of hidden layers	1, 2, 3, 4, 5
Number of units	2, 4, 8, 16, 32
Activation function	ReLU
Optimizer	Adam
Loss function	MSE
Regularization	Early stopping
Epochs	500

**Table 3 entropy-25-00071-t003:** Performance comparison of the groups.

Group	Method	SP500		CSI300	
		MAE	RMSE	MAE	RMSE
Control group	FNN	0.6110	1.1336	10.4914	15.6528
	LSTM	0.6069	1.1224	10.3923	15.4842
	GRU	0.6057	1.1220	10.3889	15.4876
	GRU-HA	0.6060	1.1209	10.3866	15.4379
High group	FNN	0.0774	0.2698	1.1830	1.9700
	LSTM	0.0312	0.0551	0.6829	1.0783
	GRU	0.0308	0.0533	0.4744	0.8126
	GRU-HA	0.0252	0.0412	0.4416	0.7841
Med group	FNN	0.1271	0.3566	3.8023	6.4337
	LSTM	0.1105	0.3324	2.4664	4.0836
	GRU	0.0671	0.2591	2.2344	3.4608
	GRU-HA	0.0121	0.1102	1.4439	2.3653
Low group	FNN	0.4972	0.6638	7.3824	9.2645
Trend group	FNN	0.1031	0.1061	0.1035	0.1192

**Table 4 entropy-25-00071-t004:** Performance comparison of ensemble results.

Methods		SP500		CSI300	
		MAE	RMSE	MAE	RMSE
Undecomposed model	FNN	1.3696	1.9580	41.2979	51.2116
	LSTM	2.1711	2.8756	73.9993	83.5591
	GRU	1.9814	2.5009	30.3358	42.6427
	GRU-HA	1.0864	1.6937	21.1347	30.0292
Decomposition-based model	CEEMD-GRU-HA	0.9373	1.3438	13.0379	17.9178
	CEGH	0.8761	1.2528	12.1849	16.8103

## Data Availability

U.S. stock market data are available in WRDS at https://wrds-www.wharton.upenn.edu/pages/about/data-vendors/nyse-trade-and-quote-taq (accessed on 10 November 2022) China stock market data are available in Wind at https://www.wind.com.cn/portal/en/WFT/index.html (accessed on 10 November 2022).

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
