# Peer review of "CEGH: A Hybrid Model Using CEEMD, Entropy, GRU, and History Attention for Intraday Stock Market Forecasting"

_entropy, 2022, doi:10.3390/e25010071_

Round 1

Reviewer 1 Report

This is a very interesting paper, with a high potential for applications to forecasting noisy time series. It is very well written, and methodologically sound. 

However, the manuscript would benefit from a discussion of how the results compare with those of the literature. There is an analysis of the results in the manuscript, but the authors do not relate their findings with those of previous literature. 

Author Response

Response to Reviewer 1 Comments

Point 1: However, the manuscript would benefit from a discussion of how the results compare with those of the literature.  There is an analysis of the results in the manuscript, but the authors do not relate their findings with those of previous literature. 

Response 1: Sorry for the confusion that the comparison with previous literature is missing. We have added more information about the comparison with previous studies in line 436 and line 468. Moreover, the baseline methods in Table 3 and Tabel 4 are referenced from the previous literature, the specific information can be found in subsection 4.2.

Reviewer 2 Report

The submitted paper is devoted to a very interesting and useful topic, 

propose a hybrid model CEGH 12 for intraday stock market forecasting,
supported by empirical study
to support the set hypothesis. Using various methods for stock market forecasting. The study has its strengths as it provides detailed information on the literature backround. Another strength is the statistical background of the paper. Moreover, I do like the consideration of meeting conditions for provided analysis presented.
To name the limitations, the strong limitation of the study is that it is based on the data coming from U.S. Stock market - it could be interesting to
compare the situation in European or other conditions (which can be the incentive for further research).

As this topic is really important to be analysed in other countries, I would recommend rewriting the methodological section and identifying methodological steps that could help other researchers replicate the research in different conditions.
I have several points which should be addressed before publishing:
1. Please, revise the references used in the paper. Some cited sources are extremely old (however relevant). But to declare the importance and topicality, the references such be mainly since 2015. I strongly recommend to rewrite the literature part of the paper and to add more Q1 and Q2 journal papers. I recommend to add also these papers:
Valaskova, K., Kliestik, T., & Gajdosikova, D. (2021). Distinctive determinants of financial indebtedness: evidence from Slovak and Czech enterprises. Equilibrium. Quarterly Journal of Economics and Economic Policy, 16(3), 639–659. https://doi.org/10.24136/eq.2021.023
Durana, P., Perkins, N., & Valaskova, K. (2021). Artificial Intelligence Data-driven Internet of Things Systems, Real-Time Advanced Analytics, and Cyber-Physical Production Networks in Sustainable Smart Manufacturing. Economics, Management, and Financial Markets, 16(1), 20–30.
Mitan, A., Siekelova, A., Rusu, M. Rovnak. M. (2021). Value-based management: A case study of Visegrad Four countries. Ekonomicko-manazerske spektrum, 15(2), 87-98.
Valaskova, K., Ward, P., & Svabova, L. (2021). Deep Learning-assisted Smart Process Planning, Cognitive Automation, and Industrial Big Data Analytics in Sustainable Cyber-Physical Production Systems, Journal of Self-Governance and Management Economics. 9(2), 9–20.
Tijani, A.A., Osagie, R.O., Afolabi, B.K. (2021). Effect of strategic alliance and partnership on the survival of MSMEs post COVID-19 pandemic, Ekonomicko-manazerske spektrum, 15(2), 126-137.
Krulicky, T., Horak, J. (2021). Business performance and financial health assessment through artificial intelligence, Ekonomicko-manazerske spektrum, 15(2), 38-51.
Kliestik, T., Nica, E., Musa, H., Poliak, M., and Mihai, E. A. (2020). Networked, Smart, and Responsive Devices in Industry 4.0 Manufacturing Systems, Economics, Management, and Financial Markets, 15(3): 23–29.
Lazaroiu, G., Kliestik, T., & Novak, A. (2021). Internet of Things Smart Devices, Industrial Artificial Intelligence, and Real-Time Sensor Networks in Sustainable Cyber-Physical Production Systems, Journal of Self-Governance and Management Economics, 9(1): 20–30.
Valaskova, K., Adamko, P., Frajtova Michalikova, K., & Macek, J. (2021). Quo Vadis, earnings management? Analysis of manipulation determinants in Central European environment. Oeconomia Copernicana, 12(3), 631–669.
Durana, P., Michalkova, L., Privara, A., Marousek, J., & Tumpach. M. (2021). Does the life cycle affect earnings management and bankruptcy? Oeconomia Copernicana, 12(2), 425–461.
2. Add information about situation in other countries and possibilities to replicate the study.
3. Provide discussion section which is missing and is a necessary part of scientific papers.

Author Response

Response to Reviewer 2 Comments

Point 1: Please, revise the references used in the paper. Some cited sources are extremely old (however relevant). But to declare the importance and topicality, the references such be mainly since 2015. I strongly recommend to rewrite the literature part of the paper and to add more Q1 and Q2 journal papers. I recommend to add also these papers.

Response 1: Thank you for the Comments. Regarding the suggestion, we have deleted three references as followings.

(1) Kumar, D.A.; Murugan, S. Performance Analysis of Indian Stock Market Index Using Neural Network Time Series Model. In Proceedings of the 2013 International Conference on Pattern Recognition, Informatics and Mobile Engineering; February 2013; pp. 72–78.

(2) Li, H.; Shen, Y.; Zhu, Y. Stock Price Prediction Using Attention-Based Multi-Input LSTM. In Proceedings of the Asian Conference on Machine Learning; PMLR, November 4 2018; pp. 454–469.

(3) Pincus, S.; Kalman, R.E. Irregularity, Volatility, Risk, and Financial Market Time Series. Proc. Natl. Acad. Sci. U.S.A. 2004, 101, 13709–13714, doi:10.1073/pnas.0405168101.

Then we have added more JCR Q1 and Q2 journal papers as the references 18, 19, 30, 41, 42.

[19] Teng, X.; Zhang, X.; Luo, Z. Multi-Scale Local Cues and Hierarchical Attention-Based LSTM for Stock Price Trend Prediction. Neurocomputing 2022, 505, 92–100, doi:10.1016/j.neucom.2022.07.016. (Q1)

[18] Zhang, Q.; Yang, L.; Zhou, F. Attention Enhanced Long Short-Term Memory Network with Multi-Source Heterogeneous Information Fusion: An Application to BGI Genomics. Information Sciences 2021, 553, 305–330, doi:10.1016/j.ins.2020.10.023. (Q1)

[30] Vințe, C.; Ausloos, M. The Cross-Sectional Intrinsic Entropy—A Comprehensive Stock Market Volatility Estimator. Entropy 2022, 24, 623, doi:10.3390/e24050623. (Q2)

[41]Lin, Y.; Lin, Z.; Liao, Y.; Li, Y.; Xu, J.; Yan, Y. Forecasting the Realized Volatility of Stock Price Index: A Hybrid [41] Model Integrating CEEMDAN and LSTM. Expert Systems with Applications 2022, 206, 117736, doi:10.1016/j.eswa.2022.117736. (Q1)

[42] Lv, P.; Shu, Y.; Xu, J.; Wu, Q. Modal Decomposition-Based Hybrid Model for Stock Index Prediction. Expert Systems with Applications 2022, 202, 117252, doi:10.1016/j.eswa.2022.117252. (Q1)

Moreover, we have already added the recommend paper in the introduction section as reference [1].

[1] Valaskova, K.; Kliestik, T.; Gajdosikova, D. Distinctive Determinants of Financial Indebtedness: Evidence from Slovak and Czech Enterprises. Equilibrium. Quarterly Journal of Economics and Economic Policy 2021, 16, 639–659, doi:10.24136/eq.2021.023.

Point 2: Add information about situation in other countries and possibilities to replicate the study.

Response 2: Thanks a lot for the constructive comments. We have reflected it in limitations and future works of the conclusion section.

Limitation (3): the experiments are only conducted on U.S. and China stock markets.

Future works (3): European or other conditions could be compared.

Moreover, we have added more information about the experimental environment settings in Experiments section to help readers replicate the study.

All the experiments are performed in Python 3.6.12. Specifically, the experiments related to Neural Network models and attention models are implemented using the deep learning end-to-end platform Tensorflow. The decomposition models are implemented using PyEMD and pyhht.

Point 3: Provide discussion section which is missing and is a necessary part of scientific papers.

Response 3: Sorry for the confusion that a discussion section is missing. The section 4 Results and Discussion has included all the discussion aspects. We discussed the finding of experiments after results analysis in the subsection 4.3 Group Results and subsection 4.4 Ensemble Results.

Reviewer 3 Report

My comments:
The topic of this paper is interesting and innovate and it will contribute in related research field. A
section of “Related Works” or “Literature Review” is necessary for this paper. Others are well presented.

Author Response

Response to Reviewer 3 Comments

Point 1: A section of “Related Works” or “Literature Review” is necessary for this paper.  

Response 1: Sorry for the confusion that a “Related Works” is missing. We have included the literature review in the section 1. Introduction from line 43 to line 125. To emphasize the “Related Works”, we have changed the first sentence in line 43 into “Several related works have been proposed to predict the stock market”. We have also added more related references to the introduction section.

[1] Valaskova, K.; Kliestik, T.; Gajdosikova, D. Distinctive Determinants of Financial Indebtedness: Evidence from Slovak and Czech Enterprises. Equilibrium. Quarterly Journal of Economics and Economic Policy 2021, 16, 639–659, doi:10.24136/eq.2021.023.

[18] Zhang, Q.; Yang, L.; Zhou, F. Attention Enhanced Long Short-Term Memory Network with Multi-Source Heterogeneous Information Fusion: An Application to BGI Genomics. Information Sciences 2021, 553, 305–330, doi:10.1016/j.ins.2020.10.023.

[19] Teng, X.; Zhang, X.; Luo, Z. Multi-Scale Local Cues and Hierarchical Attention-Based LSTM for Stock Price Trend Prediction. Neurocomputing 2022, 505, 92–100, doi:10.1016/j.neucom.2022.07.016.

[30] Vințe, C.; Ausloos, M. The Cross-Sectional Intrinsic Entropy—A Comprehensive Stock Market Volatility Estimator. Entropy 2022, 24, 623, doi:10.3390/e24050623.

Lin, Y.; Lin, Z.; Liao, Y.; Li, Y.; Xu, J.; Yan, Y. Forecasting the Realized Volatility of Stock Price Index: A Hybrid [41] Model Integrating CEEMDAN and LSTM. Expert Systems with Applications 2022, 206, 117736, doi:10.1016/j.eswa.2022.117736.

[42] Lv, P.; Shu, Y.; Xu, J.; Wu, Q. Modal Decomposition-Based Hybrid Model for Stock Index Prediction. Expert Systems with Applications 2022, 202, 117252, doi:10.1016/j.eswa.2022.117252.